# Image-Based Pothole Detection Using Multi-Scale Feature Network and Risk Assessment

**Dong-Hoe Heo** [1] , **Ji-Yoon Choi** [1] , **Sang-Baeg Kim** [2] , **Tae-Oh Tak** [1,*] **and Sheng-Peng Zhang** [1,*]

[1] Department of Mechanical and Biomedical Engineering, Kangwon National University, Chuncheon City 24341, Gangwon-do, Republic of Korea
[2] Kai Networks Corporation, Suwon City 16463, Gyoenggi-do, Republic of Korea
[*] Correspondence: totak@kangwon.ac.kr (T.-O.T.); zsp363527125@gmail.com (S.-P.Z.)

**Abstract:** Potholes on road surfaces pose a serious hazard to vehicles and passengers due to the difficulty detecting them and the short response time. Therefore, many government agencies are applying various pothole-detection algorithms for road maintenance. However, current methods based on object detection are unclear in terms of real-time detection when using low-spec hardware systems. In this study, the SPFPN-YOLOv4 tiny was developed by combining spatial pyramid pooling and feature pyramid network with CSPDarknet53-tiny. A total of 2665 datasets were obtained via data augmentation, such as gamma regulation, horizontal flip, and scaling to compensate for the lack of data, and were divided into training, validation, and test of 70%, 20%, and 10% ratios, respectively. As a result of the comparison of YOLOv2, YOLOv3, YOLOv4 tiny, and SPFPN-YOLOv4 tiny, the SPFPN-YOLOv4 tiny showed approximately 2–5% performance improvement in the mean average precision (intersection over union = 0.5). In addition, the risk assessment based on the proposed SPFPN-YOLOv4 tiny was calculated by comparing the tire contact patch size with pothole size by applying the pinhole camera and distance estimation equation. In conclusion, we developed an end-to-end algorithm that can detect potholes and classify the risks in real-time using 2D pothole images.

**Keywords:** pothole; 2D object detection; convolutional neural network; multi-scale detection; risk assessment

## 1. Introduction

### 1.1. Problem Definition and Motivation

According to a press release by the Ministry of Road Transport & Highways—"Road accidents in India—2021"—[1], a total of 7189 pothole-related accidents, with 2952 deaths and 6167 injuries, occurred from 2020 to 2021, posing a significant threat to the road users. Furthermore, the cost that the governments must cover for the loss of human lives and road accidents resulting from potholes is rapidly increasing [2]. Recently, potholes have been increasing due to the more severe weather brought on by global warming [3]. Accordingly, government agencies in each country ought to maintain road surfaces in good condition to guarantee health and safety; hence the timely detection of asphalt cracks and potholes is essential. In particular, various pothole detection methods have been introduced as part of road data collection for road surface maintenance. Methods for detecting potholes [4] include 3D reconstruction, vibration, and 2D image-based pothole detection.

The 3D image reconstruction method can analyze the depth and width of the pothole by converting the road surface into a three-dimensional image using stereo vision-based camera and laser scanner [5,6]. However, developing a 3D sensor system is expensive and very sensitive to vibrations transmitted from road surfaces to vehicles, resulting in low accuracy [7].

On the other hand, the vibration sensor-based method can estimate the depth of a pothole and detect its position by sensing its impact or motion using an acceleration sensor [8]. However, this method cannot predict potholes in advance because it detects

potholes only when the vehicle passes directly through the road surface. In addition, potholes cannot be distinguished when they pass through manholes or speed bumps.

The 2D Image-based detection method recognizes the textures and characteristics of potholes based on images of the road surface acquired using a camera [9]. This method helps prevent accidents in advance and facilitates real-time detection because potholes can be identified from a relatively long distance. In addition, it can be used with a camera sensor, which is much less expensive than the laser scanner used in the 3D method. Therefore, various 2D image-based detection methods that can operate with high efficiency and low cost have been developed [10].

Moreover, with the rapid development of deep learning technology, convolutional neural networks (CNNs) that can automatically learn object features from images have been introduced. The object-detection method classifies the types of objects to be detected and draws a square-shaped bounding box around the classified objects to indicate their locations. Among the various object detection techniques, the You Only Look at Once (YOLO) algorithm has the fastest computational speed among existing deep-learning-based object detection techniques, and enables real-time detection. However, real-time object detection cannot be guaranteed in embedded systems with low-performance GPUs and CPUs, even if the computation speed of YOLO is fast [11]. Therefore, the YOLO-tiny series algorithm, advantageous for real-time detection in embedded systems, has been widely applied to pothole detection [12]. However, YOLO-tiny series algorithms have lower accuracy than general YOLO algorithms because the network size is reduced for computation speed.

Not only that the importance of pothole detection, but it is also necessary to assess the risk for potholes of various sizes and shapes because potholes force different impacts on tires and suspensions depending on their size [13]. Therefore, many researchers and experts have recently conducted studies to evaluate the risk of potholes based on their size and volume [14]. Typical techniques for assessing the severity of potholes include size-based methods [15], and deep learning-based methods [16]. The size-based pothole risk assessment method evaluates risk by comparing depth, volume, area, and size with predefined classification criteria. However, most risk classification conditions used to compare sizes are highly subjective depending on the country, experts, and even researchers. Deep learning-based pothole risk assessment methods classify risk into low, medium, and high stages using data acquired by sensors [17]. However, this method is hard to use in a real-time application, and it is even more challenging to know the risk classification criteria trained by the deep neural network.

### 1.2. Contributions

To address the above problem, we developed the SPFPN-YOLOv4 tiny algorithm, which has a higher accuracy than the general YOLO algorithm, and enables real-time detection by improving the existing YOLOv4 tiny. Spatial pyramid pooling (SPP) and feature pyramid network (FPN) are added to CSPDarknet53-tiny, which is the backbone of YOLOv4 tiny. In the stage of detection performance comparison, SPFPN-YOLOv4 tiny achieved the highest performance in the mean average precision (mAP) (intersection over union (IoU = 0.5)), F1-score, and frames per second (FPS). Additionally, previous studies have reported that it is crucial not only to detect potholes, but also to judge their severity according to their size, as potholes have a negative impact on suspension, tire damage, and drivers depending on their size [13]. Gupta et al. [18] conducted a deep neural network study using thermal image images and proposed the need for image-based severity classification as an agenda for future research. In addition, previously studied pothole detection algorithms only focus on detecting potholes on the road surface and cannot distinguish large potholes potentially affecting traffic accidents. To overcome the limitations of existing detection algorithms, we present a novel risk assessment algorithm that can be practically adapted to drivers and passengers with visualization signals. This risk assessment algorithm classifies risk into three classes of severity: danger, safety, and

caution, by comparing the tire footprint size grounded on the road with the actual pothole size calculated using the pinhole camera and distance equations. The key contributions of this study are as follows:

1.  Assuming a low-spec hardware system, we developed a lightweight pothole detection algorithm capable of real-time detection without a GPU.
2.  By applying data augmentation techniques, such as gamma regulation, horizontal flip, and scaling, we compensate for the deficient dataset and increase the adaptation of deep learning models in various environments (i.e., overexposed, underexposed, unpaved, asphalt pavement)
3.  We proposed a new risk assessment criterion to compare the size of potholes with the tire contact patch size to identify large potholes that are likely to affect traffic accidents.

The remainder of this paper is organized as follows: Section 2 presents a study of detection algorithms and risk assessment associated with potholes. Section 3 presents the data preprocessing, SPFPN-YOLOv4 tiny, and the risk assessment algorithm applied in this study. Section 4 presents the results of the object detection performance comparison and the validation of the risk assessment algorithm. Section 5 presents conclusions, limitations, and future research objectives.

## 2. Related Work

### 2.1. Pothole Object Detection

Early 2D pothole detection techniques were machine-learning-based classification algorithms. A machine learning-based pothole detection algorithm distinguished the road surface into defective and non-defective areas by setting a feature classifier subjectively defined by the user [19]. These shape-based feature classifiers often detect the shaded part of the road surface as a pothole. Powell and Sateshjumar [20] detected potholes by considering its shape to be approximately elliptical using a machine learning-based pothole detection algorithm combined with shadow removal algorithms. However, machine-learning-based pothole detection algorithms can only classify potholes from images and cannot display their locations.

As deep learning technology has developed rapidly, these machine-learning-based limitations have been overcome with the development of object detection techniques in computer vision. These object detection techniques are classified mainly into two-stage detectors and one-stage detectors. The two-stage detector extracts numerous candidates in the area where the object exists and sequentially processes the classification and localization processes; thus, the accuracy is high, but the computation speed is slow. However, as the one-stage detector processes the classification and localization processes simultaneously, the accuracy is low, but the computation speed is fast. YOLO has the fastest computation speed among existing one-stage detector-type object detection models, and most researchers are currently developing YOLO-based 2D pothole detection algorithms.

Ukhwah et al. attached a camera for road surface detection to the rear of a vehicle to capture road surface images in East Java Province [21] and trained YOLOv3 [22], YOLOv3 tiny [23], and YOLOv3 SPP [24] using selected 448 images. As a result of training various YOLO models, each model achieved a mAP of 83.43%, 79.33%, and 88.93%, respectively. Through this result, the author argued that the YOLOv3 SPP combined with SPP achieved the highest accuracy. Park et al. [25] split a 665 pothole image dataset into training, validation, and test datasets by selecting 70%, 20%, and 10% of the total sample, respectively, then trained 598 pothole image datasets to YOLOv4 [26], YOLOv4 tiny [27], and YOLO v5s [28] models. As a result of the performance evaluation of the model, YOLOv4, YOLOv4 tiny, and YOLOv5s achieved 77.7%, 78.7%, and 74.8% mAP(IoU = 0.5), respectively. In conclusion, the author noted that YOLOv4 tiny achieved the highest accuracy, and there is a need to extend the backbone network of YOLOv4 tiny to improve accuracy. Bucko et al. [2] trained 1052 images collected differently under clear, rainy, sunset, evening, and night weather conditions. Especially

under clear weather conditions, YOLOv3, YOLOv3 SPP, and Sparse R-CNN achieved 77%, 79.1%, and 72.6% mAP (IoU = 0.5), respectively. Dharnesshkar et al. [29] trained a dataset of 1500 Indian road images collected from Coimbatore, Idukki, and Kymily on the YOLOv2, YOLOv3, and YOLOv3 tiny models, achieving mAP(IoU = 0.5) 45.33%, 38.41%, and 49.71%, respectively. Asad et al. [30] trained 665 pothole datasets on the YOLOv4 tiny, YOLOv4, and YOLOv5 models, and verified the real-time detection possibility in a low-end embedded system. An OAK-D camera with a single main board (Raspberry Pi) was used to detect potholes. One of the limitations was that only a YOLOv4 tiny model achieved real-time detection of 31.76 FPS among various YOLO models.

However, although the existing computation speed of the YOLOv4 tiny model is fast, this model needs to extract richer object features to be detected. Therefore, in this study, we attempted to improve the accuracy by combining multi-scale feature networks, such as SPP and FPN, with the YOLOv4 tiny architecture to extract rich object features. The multi-scale feature networks are used to improve the detection performance. SPP extracts pothole features of various sizes by applying multiple max-pool filters to compensate for the spatial information loss of the image that occurs during max-pooling, which is a part of the convolutional neural network process [31]. The FPN compensates for spatial information loss by combining high-level feature maps generated in deep convolutional layers with low-level features generated in shallow convolutional layers [32]. This mechanism of combining various feature maps makes greater use of the spatial information of the image by extracting the rich features of the object, thereby improving the overall performance of the deep learning model. Table 1 summarizes selected studies on pothole detection and shows their contribution, models, hardware systems, FPS, and mAP (IoU = 0.5).

**Table 1.** A summary of recent road pothole detection algorithm performance.

| References & Year | Contribution | Model | CPU | GPU | FPS | mAP(IoU@0.5) |
|---|---|---|---|---|---|---|
| Ukhwah et al. [21], 2019 | Detection of potholes and area estimation | YOLOv3 YOLOv3 tiny YOLOv3 SPP | Intel ® Xeon (R) CPU@2.30GHz | Tesla T4 (13 GB) | 0.4 | 83.43% 79.33% 88.93% |
| Park et al. [25], 2021 | Comparison with various YOLO models | YOLOv4 YOLOv4 tiny YOLOv5s | – | Tesla K80 (12 GB) | – | 77.7% 78.7% 74.8% |
| Asad et al. [30], 2022 | Detection of pothole using Raspberry Pi 4 | YOLOv2 YOLOv3 YOLOv4 YOLOv4 tiny YOLOv5 | ARM Cortex-A53@1.4GHz | – | 3.20 2.39 1.98 31.76 18.25 | 81.21% 83.60% 85.48% 80.04% 95.00% |
| Bucko et al. [2], 2022 | Detection potholes under adverse weather condition | Sparse R-CNN YOLOv3 YOLOv3-SPP | – | – | – 28.57 27.78 | 72.6% 77.1% 79.1% |
| Dharnesshkar et al. [29], 2020 | Detection potholes of the India road | YOLOv2 YOLOv3 YOLOv3 tiny | – | GeForce GTX 1060 | – | 45.33% 38.41% 49.71% |

### 2.2. Risk Assessment Methods

Due to potholes impacting vehicles differently depending on their size, government agencies in the U.S. and China have set standards to classify the risk of potholes according to their depth and area using acceleration sensors, stereo vision cameras, and laser sensors. China's Highway Performance Assessment Standard (HPAS) states that if a pothole area is less than 0.1 m$^2$, the risk is low, and if the area is greater than 0.1 m$^2$, the risk is high. The Long-term Pavement Performance Program (LTTP) in the United States defines pothole depth as low-risk when the depth is less than 25 mm, medium-risk when the depth is greater than 25 mm and less than 50 mm, and high-risk when the depth is greater than 50 mm. According to a press release by the Royal Automobile Club (RAC) Foundation [33], Local Highway Authorities (LHA) in the United Kingdom applies a "risk-based" road maintenance method that evaluates the risk depending on the width and depth of the pothole. Solanke et al. [14] distributed questionnaires to road-related experts to classify risk levels to determine the severity of

potholes for road surface maintenance. However, this type of pothole risk assessment method is subjective and wastes human resources when measuring the pothole size. In addition, this is not practical because the size cannot be measured when the vehicle is driving at speed. Table 2 summarizes the risk classification criteria and conditions of potholes identified in the LTTP (United States) and HPAS (China) [5].

**Table 2.** Pothole risk classification specified in LTTP and HPAS.

| Property | Degree of Risk | LTTP (USA) | HPAS (China) |
|---|---|---|---|
| Depth | Low | Below 25 mm | – |
| | Moderate | Between 25 and 50 mm | |
| | High | Greater than 50 mm | |
| Area | Low | – | Below 0.1 m$^2$ |
| | High | | Greater than 0.1 m$^2$ |

According to the literature study above, the pothole size significantly correlates with severity of the road. If the pothole detection algorithm is applied to risk assessment, the size of the pothole can be measured in real-time, which can be used for road maintenance and quickly recognized in driving situations. In Korea, no criteria exist for evaluating pothole risk, but a few studies have been conducted. Chung-Hyun et al. [34] attached an acceleration sensor to a vehicle. They presented a risk index divided into four stages: Caution, Alert, Warning, and Danger, according to the Z-axis acceleration value generated when passing through the pothole. However, this method evaluates the degree of risk based on a sensor that detects vibrations and can only detect potential hazards when the vehicle directly passes through the road surface with the potholes. Therefore, this method has disadvantages because it cannot recognize dangerous potholes in advance.

In contrast, applying a risk assessment requires using 2D pothole images that can predict dangerous potholes before a vehicle passes through the pothole. Kortmann et al. [35] performed severity classification using deep learning based on 2D road images to establish a route plan for an autonomous vehicle. The author classified road hazards into three stages: low, medium, and high, depending on the road features, potholes, and depths of the road surface. According to this risk assessment, road hazards were classified as 'high' risk even if there were small potholes on the road that could not affect the vehicle. Therefore, this study presents a new 2D pothole risk assessment that compares the actual pothole size and tire contact patch area grounded on a road surface to address the limitations of previous pothole risk assessment studies.

## 3. Methods

This study developed the SPFPN-YOLOv4 tiny model by combining multi-scale feature networks, such as SPP and FPN, with the YOLOv4 tiny model suitable for embedded systems. A new 2D pothole risk assessment standard was proposed to visually indicate risk signals to the driver by comparing the size of the pothole detected using the developed model with the size of the tire contact patch area, as shown in Figure 1.

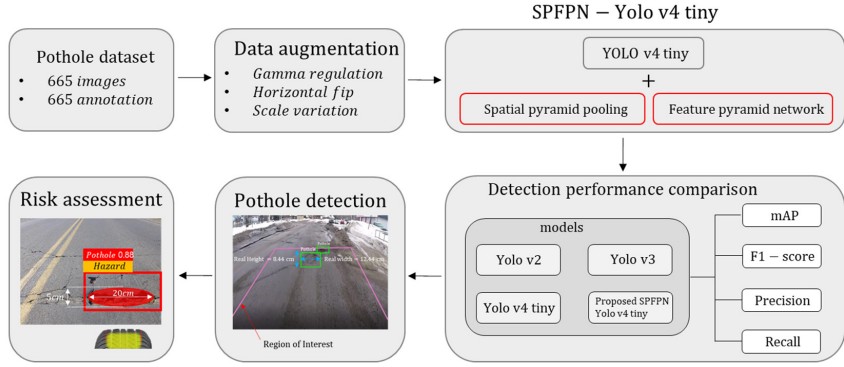

**Figure 1.** Research and risk assessment procedure using SPFPN YOLOv4 tiny network.

First, a pothole dataset comprising 665 images and annotations was used to train various deep-learning models. Data augmentation techniques, such as gamma regulation, scaling, and horizontal flip, were randomly applied to the original data to obtain 2665 datasets to compensate for the insufficient dataset. Subsequently, 2400 training and validation datasets were used to train the model, and the remaining 265 test datasets were used for model performance evaluation after the training was completed. To verify the performance of the SPFPN-YOLO v4 tiny developed in this study, YOLOv2, YOLOv3, and YOLOv4 tiny were selected as benchmarks, and the improvement in accuracy was verified by measuring the mAP(IoU = 0.5), precision, recall, and F1-score for each model. The FPS values of the models were compared without a GPU to confirm real-time detection, even on low-end hardware.

In addition, the bounding box of the detected pothole was converted to the actual size using the pinhole camera and distance estimation equations to determine the risk according to the relationship between the pothole and tire contact patch size. Risk evaluation criteria categorized the potholes into three stages: hazard if they were 1.2 times larger than the tire contact patch size, safety if they were 1.2 times smaller than the tire contact patch size, and caution if they were the same size.

### 3.1. Dataset Preprocessing

3.1.1. Dataset Preparation and Data Augmentation

The dataset comprises 665 image and annotation files in the Pascal VOC format provided by Kaggle [36]. The dataset's images were captured under various environmental conditions, such as public roads, unpaved roads, and potholes, including rainwater. Ground truth information in Pascal VOC format was converted into a YOLO annotation format. The supervised deep learning model depends on the quality and quantity of the dataset; therefore, data augmentation, such as horizontal flip, scaling, and gamma regulation was applied to the original image datasets so that insufficient datasets and models could adapt strongly to low-quality photos, light reflection, overexposed, and underexposed data, as shown in Figure 2. The scale was multiplied by 0.7 for each pixel size (width, height) in the image, gamma regulation was randomly multiplied by 0.5–2.0 for each channel in the image, and the image completely flipped horizontally. After image transformation, the annotation files of the dataset were transformed. As shown in Figure 2, the data augmentation technique for the ground truth was applied only to images with horizontal flip and scaling, where objects were deformed, and not to images with gamma regulation, which did not change the shape of the object. The original ground truth is marked in green, and the ground truth to which each data augmentation was applied is marked in red. Subsequently, 2665 datasets with augmented techniques were divided into training, validation, and test datasets at ratios of 70%, 20%, and 10%, respectively. Two thousand four hundred datasets were used for model training, and 265 test datasets were used for test image inference. As potholes mainly cause serious incidents in highway environments where vehicles drive at high speeds [37], we manually picked 50 pothole images in the asphalt pavement environment similar to the highway road when measuring the Precision, Recall, F1-score, and FPS after training the model.

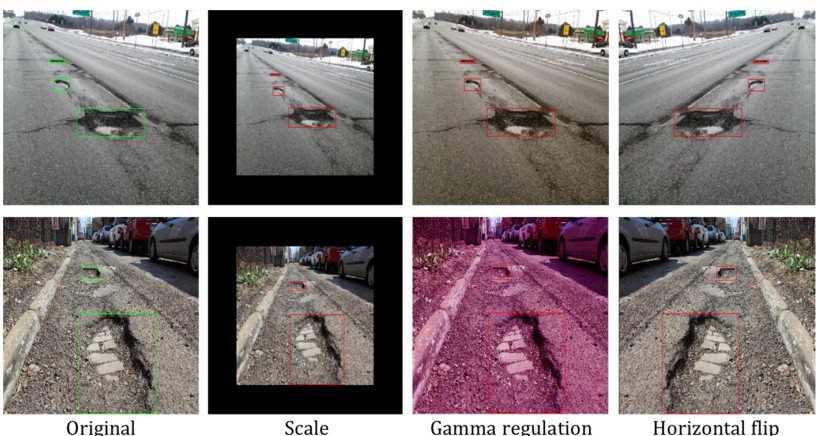

| Original | Scale | Gamma regulation | Horizontal flip |

**Figure 2.** Pothole images subset after data augmentation.

### 3.1.2. Dataset Configuration

As shown in Figure 3, the total number of datasets after data augmentation was applied was 2665, divided into 1865, 535, and 265 images for the training, validation, and test data, respectively. Subsequently, additional dataset splits were performed. In order to prevent the deep learning model from overfitting to the original data set, it was first distributed to 60%, 20%, and 20%, respectively, then redistributed to 60%, 18%, and 22%, respectively, by moving a small amount of data. Additionally, for improving model's detection accuracy under challenging condition, different ratios were applied to the data set to which the data augmentation method was applied and divided into training, validation, and test data, as shown in Figure 3. The scaling-applied dataset is a data augmentation method to train small potholes well, divided into 540, 120, and 40 data for the train, valid, and test datasets. The Gamma regulation-applied dataset is a method for precisely detecting potholes in a noisy environment, divided into 415, 145, and 40 data for the train, valid, and test datasets. The Horizontal flip-applied dataset is intended to compensate for the deficient dataset, divided into 510, 150, and 40 data for the train, valid, and test datasets. Since the data augmentation-applied dataset was applied to improve the detection performance of the model under an adverse environment and prevent overfitting, it was allocated to 40 sheets for the test dataset.

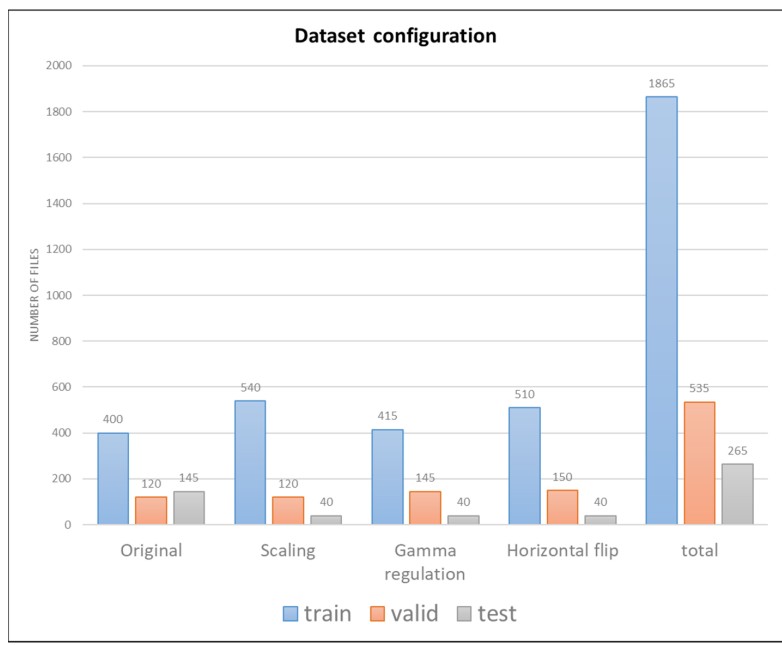

**Figure 3.** Pothole dataset configuration after data augmentation.

### 3.2. Development SPFPN-YOLOv4 Tiny

This section is divided into two sections. Section 3.2.1 describes the importance and functionality of detection algorithms based on the multi-scale feature network used for developing of the SPFPN-YOLOv4 tiny proposed in this work. Additionally, we show how the applied SPP and FPN are coupled to the existing YOLOv4 tiny together in Figure 4. Section 3.2.2 describes K-means++ clustering for efficiently applying of YOLO's detection mechanism, anchor box, as the network changes.

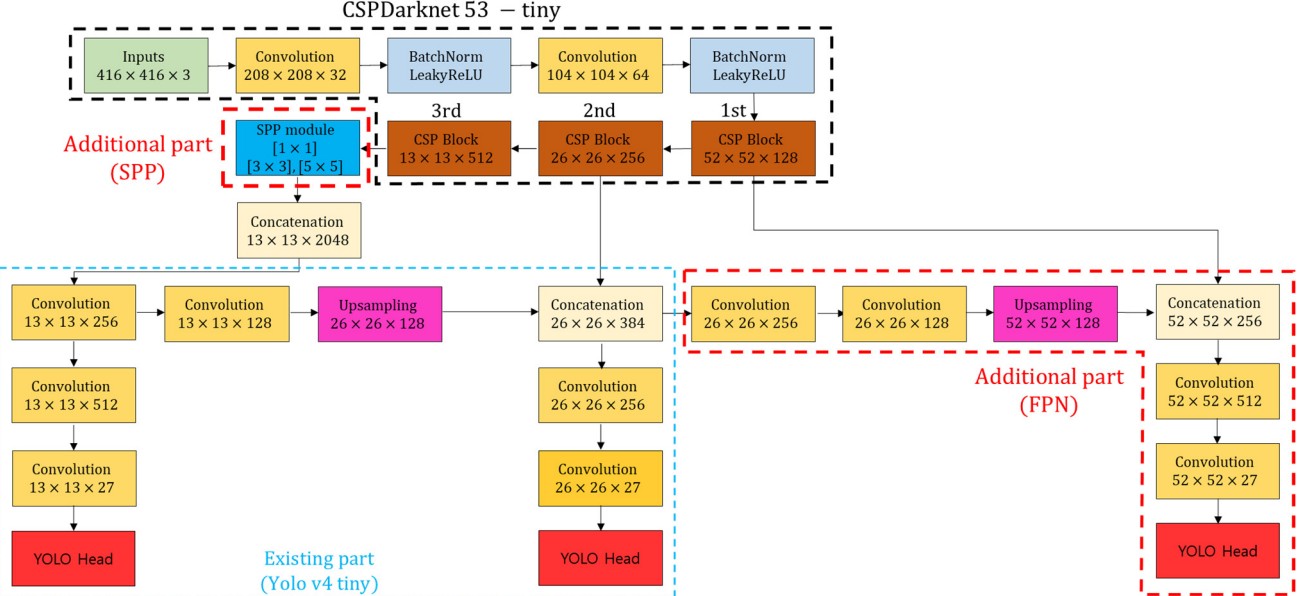

**Figure 4.** SPFPN YOLOv4 tiny overall architecture.

### 3.2.1. Multi-Scale Feature Network

The performance of the object detection model depends on the amount of core information about the objects in the convolutional neural network. Multi-scale feature networks are CNN-based structures that allow the independent detection of feature maps of different sizes fusing between high-level and low-level feature maps to create a new one with more information on objects. This method is an important area of research, in which various multi-scale object detection methods have been introduced to achieve better performance in recent years. Object detection based on multi-scale detection is mainly divided into (1) independent detection of several feature maps extracted from different layers of the network, and (2) fusion of several feature maps extracted from different layers of the network.

In case (1), the data was first applied to a single-shot detection network (SSD) for object detection in independent feature maps generated from different layers of CNN, and detecting small objects from feature maps combined with high-level and low-level recorded better accuracy than detecting small objects from feature maps extracted from low-level [38]. In case (2), Li et al. [39] first reported that state-of-the-art performance in pedestrian detection using scale-aware mechanisms was obtained by combining and adding output feature maps using large and small sub-networks for the fusion of several feature maps. He et al. [31] increased the detection performance by applying several max-pool filters and SPP, which can create feature maps regardless of the input image size. Lin et al. [32] developed FPN and a top-down side connection structure integrate information between high-level and low-level feature maps by combining local area features of different sizes.

This literature review demonstrated that multi-scale feature networks are effective in detecting small objects using even detection in independent feature maps and achieve high accuracy when using feature fusion methods by connecting high-level and low-level feature maps. Therefore, in this study, we combined the backbone network of the YOLOv4

tiny model with SPP and FPN, which are part of a multi-scale feature network type, to improve detection performance and detect small potholes barely detected in asphalt roads.

### 3.2.2. SPFPN YOLOv4 Tiny

Several researchers and road maintenance organizations have recently adopted the YOLO model, which has the fastest computation speed and highest accuracy among the one-stage detector-type object detection models. YOLO detects and predicts by selecting the prediction bounding box with the highest confidence score by dividing the input image into an N × N grid cell. In addition, YOLOv4 tiny is the simplified version of YOLOv4, which showed a high performance of 43.5% average precision (AP) on the MS COCO data sets [40]. YOLOv4 has a smaller CNN structure than YOLOv4; therefore, the detection speed is fast, but with lower accuracy. Even though the YOLO-tiny series showed low accuracy, they can achieve real-time detection even in low-spec embedded systems, such as cameras and Raspberry Pi, which lack hardware specifications, unlike ordinary computers. Therefore, YOLOv4 tiny was adopted in this study because it is a highly suitable deep-learning model for detecting potholes while attaching a camera to the vehicle.

YOLOv4 tiny detects objects based on CSPDarknet53-tiny, which applies CSPNet [41] and one FPN, which is a type of multi-scale network [27]. Figure 4 shows the overall structure of the SPFPN-YOLOv4 developed in this study. The black dotted line is CSPDarknet53-tiny which is the backbone of YOLOv4 and consists of three CSP blocks and two convolutional layers. The area marked by the blue dotted line represents the original YOLOv4 tiny head, and the object is finally detected in a feature map with sizes of 13 × 13 and 26 × 26. The FPN consisting of an upsampling layer, and a convolutional layer was used to connect the 13 × 13 feature maps and 26 × 26 feature maps.

Furthermore, 13 × 13 feature maps are high-level feature maps containing the most intensive feature information of objects, and 26 × 26 maps can be considered middle-level feature maps with less intensive spatial information than the 13 × 13 feature maps. As the size of the feature map decreased, the shape of the smaller object became more distorted, making it difficult to detect small potholes. Therefore, we added FPN to improve the small object detection accuracy by utilizing low-level feature maps of 52 × 52 size. Moreover, an SPP network is added to the third CSP block, which has a high probability of creating small distorted objects. The SPP network divides image sizes into various images by applying max-pool filters, then merges them, resulting in less information loss than the conventional max-pooling process. Additionally, [1 × 1], [3 × 3], and [5 × 5] max-pool filters are applied in added SPP network, and a total of 13 × 13 × 2048 feature maps are generated after passing this network. The feature maps passing the third CSP block connected to the SPP are first connected to the high-level feature map of 13 × 13, and the high-level feature map is sequentially connected to the middle- and low-level feature maps through two FPN. Finally, 52 × 52 low-level feature maps responsible for small object features are connected with 13 × 13 and 26 × 26 feature maps that train low-level features, such as edges, shapes, and textures, and reflect intensive features, such as key information representative of potholes.

### 3.3. K-Means++ Clustering

An anchor box has a high probability of a previously defined object being present. The YOLO object detection method draws an accurately predicted bounding box by dividing the image into a regular grid and calculating the error between the ground truth and predefined anchor boxes. The number of objects that can be detected in one grid is the same as the number of anchor boxes defined in advance, and the detection accuracy is high when the size of the anchor box is similar to that of the ground truth box. Therefore, K-means ++ clustering [42] was applied to improve the IoU score between the prediction bounding box and the ground truth box. The K-means++ clustering method was applied as follows:

(1) One of the data points was randomly selected and designated as the first center point.

(2) The distance to the first center point for the remaining points was calculated.

(3) A data point is placed as far away as possible from the defined center point and then designated as the next center point.

(4) Steps 2 and 3 are repeated until there are K centers, using the following equation:

$$d = max_{(j:1 \to m)} \| x_i - centroid_j \|^2 \qquad (1)$$

where $x_i$ is a point in the dataset, $centroid_j$ is any data selected as the centroid, d is the longest distance between $x_i$ and $centroid_j$, and m is the total number of centroids K. Focusing on the K-means ++ clustering mechanism, the farthest distance between centroids as K and points were calculated using the following equation:

$$d(\text{Ground truth, centroid}) = 1 - IoU(Ground\ truth,\ centroid) \qquad (2)$$

The IoU was calculated by dividing the overlapping area between the anchor box and ground truth box by the sum of the areas of the two boxes. The lower the IoU value, the farthest distance between the centroids and the point is calculated; therefore, it is possible to set the center point farthest from the point. Anchor box values applied with K-means ++ clustering are [35, 21], [67, 44], [116, 64], [125, 126], [182, 210], [130, 110], [110, 60], [72, 48] and [40, 22]. By setting the number of anchor boxes to 9, an IoU score of 71.02% was achieved.

### 3.4. Object Detection Performance Evaluation Metrics

For the performance evaluation of SPFPN-YOLOv4tiny developed in this study, AP, precision or mAP, recall, and F1-score calculated using confusion matrix index, which is widely used as a computer vision performance evaluation technique, was used as an evaluation index. Table 3 defines the confusion matrix index as follows [25].

**Table 3.** Confusion matrix index.

| Actual ╲ Prediction | Predicted as Positive | Predicted as Negative |
|---|---|---|
| Positive | True Positive (TP) | False Negative (FN) |
| Negative | False Positive (FP) | True Negative (TN) |

As shown in Table 3, a true positive (TP) is the correct detection of the model. A false positive is the object's false detection, which does not exist in the image. Precision is defined as the ratio of the number of true positives (TP) among (TP + FP), and recall is the ratio of true positives (TP) out of those that are positive (TP + FN), which can be calculated using Equations (3) and (4). Both precision and recall values are affected by IoU, which refers to the overlapping ratio between the prediction bounding box and ground truth, as shown in Equation (5).

$$\text{Precision} = \frac{TP}{TP + FP} \qquad (3)$$

$$\text{Recall} = \frac{TP}{TP + FN} \qquad (4)$$

$$\text{IoU} = \frac{A^P \cap A^{GT}}{A^P \cup A^{GT}} \qquad (5)$$

where $A^P$ is the area of the prediction bounding box, and $A^{GT}$ is the area of the ground truth box. With these precision and recall variables, the F1-score and AP can be calculated using Equations (3) and (4). The F1-score is the harmonic average of the precision and recall. The AP is the weighted sum of the precedence at each IoU threshold, and the weight

at this time is calculated as the difference between the recall value at the previous threshold and the current threshold.

$$\text{F1 score} = 2 \times \frac{\text{Precision} \times \text{Recall}}{\text{Precision} + \text{Recall}} \tag{6}$$

$$\text{AP@IoU} = \sum_{i=1}^{M} (R_n - R_{n-1})(P_n) \tag{7}$$

$$\text{mAP@IoU} = \frac{1}{\text{N}} \sum_{i=1}^{N} AP_i \tag{8}$$

The mAP is the average AP value for each class among the indices used to evaluate object detection performance, but it is regarded as the same as mAP because only one class was used in this study. AP and mAP can evaluate the reliability of the model by setting the IoU threshold. The IoU threshold when measuring the model performance evaluation was set to 0.5. In the training phase performance evaluation, mAP(IoU = 0.5) was applied, and precision, recall, and F1-score were applied to model performance evaluation for 50 images in the test dataset. In addition, FPS was also applied to measure the reference speed of the deep learning model.

### 3.5. Risk Assessment

#### 3.5.1. Pothole Distance Estimation

Before calculating the actual pothole size using 2D images, the distance between the camera and pothole was calculated by assuming a monocular camera based on the distance estimation equation, as shown in Figure 5 [43]. The prerequisites for this method are as follows: (1) the detected object is attached to the ground, and (2) the distance is estimated using only the detected pothole. The distance between the detected pothole and the vehicle with a camera attached can only be determined under the situation where these two conditions are satisfied.

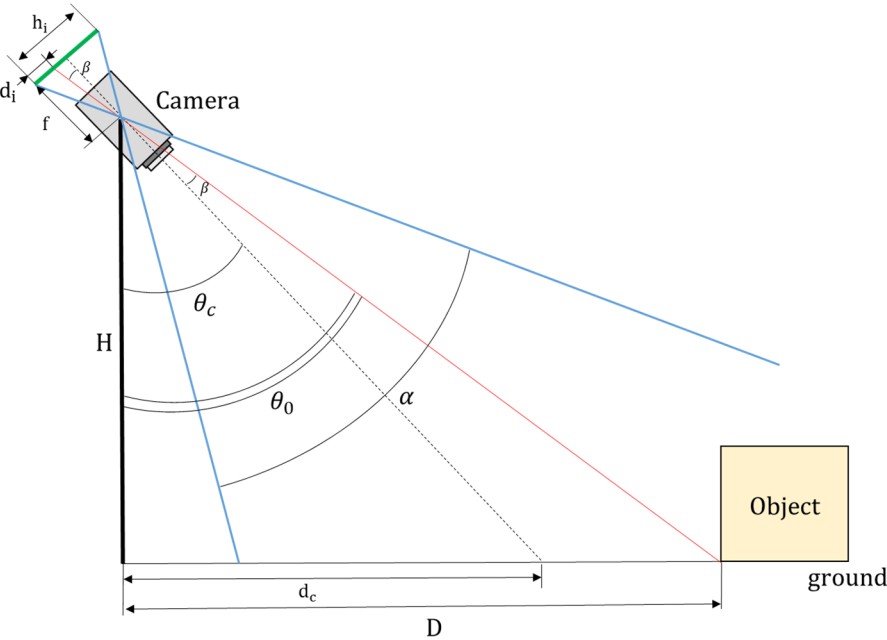

**Figure 5.** Distance estimation between object and camera.

When a pothole enters the camera's field of view, it is detected with a prediction bounding box using an object-detection algorithm. Assuming that the pothole is in contact

with the ground, the horizontal distance can be expressed using Equation (9) by applying the trigonometric ratio.

$$D{\cdot}\tan(\theta_0) = \text{H}{\cdot}\tan(\theta_c + \beta) \tag{9}$$

where H is the height from the ground to the camera lens; $\theta_c$ is the angle of inclination from the vertical axis pixel height of the image; $\beta$ is the angle from the camera to the detected pothole; and $\theta_0$ is the angle from the vertical axis to the pothole. $\theta_0$ can be calculated by following Equation (10):

$$\tan(\theta_0) = \frac{D}{H} \tag{10}$$

where $d_c$ is the horizontal distance between the actual position and camera. $\beta$ can be calculated using Equation (11):

$$\tan(\beta) = \frac{\frac{h_i}{2} - d_i}{f} \tag{11}$$

where $h_i$ is the pixel height of the image, $d_i$ is the horizontal distance from the actual position to the camera, and f is the focal length of the lens, which can be calculated using Equation (12):

$$\text{f} = \frac{h_i}{2{\cdot}\tan\left(\frac{\alpha}{2}\right)} \tag{12}$$

where $\alpha$ is the horizontal viewing angle of the lens. Finally, substituting all the variables obtained through Equations (10)–(12), Equation (10) can be expressed as in Equation (13):

$$\text{D} = \text{H}{\cdot}\left[\tan\left(\theta_c + \tan^{-1}\left(\frac{\frac{h_i}{2} - d_i}{2{\cdot}\tan\frac{\alpha}{2}}\right)\right) - \tan\left(\theta_c - \frac{\alpha}{2}\right)\right] \tag{13}$$

3.5.2. Pinhole Camera Model

Figure 6 illustrates the principle of the pinhole camera and shows the image plane projected through the pinhole of the lens. The pinhole is a simplified camera model that projects an external image onto an image plane and considers image pixels and absolute coordinates. In Figure 6, the axis passing through the hole of the lens is called the optical axis, and the distance from the hole to the location where the image on the back is formed is called the focal length. The focal length calibrated for each of the *x*- and *y*-axes of the camera lens are called the *x*-axis focal length ($f_x$) and *y*-axis focal length ($f_y$), respectively. The central point in the image plane on which the image is projected is called the principal point ($C_x, C_y$). The pinhole camera parameters are expressed as a camera matrix in Equation (14):

$$\text{D} = \lambda \begin{bmatrix} u \\ v \\ 1 \end{bmatrix} = \begin{bmatrix} f_x & \gamma & c_x & 0 \\ 0 & f_y & c_y & 0 \\ 0 & 0 & 1 & 0 \end{bmatrix} \begin{bmatrix} x_W \\ y_W \\ Z_W \\ 1 \end{bmatrix} \tag{14}$$

where u, v are the pixel coordinates in which the image is projected onto the image plane; $(x_w, y_w, z_w)$ are the 3D coordinates of the detected pothole; $(c_x, c_y)$ are the principal point; $\gamma$ is the distortion rate, which is set to 0, assuming that there is no distortion rate; $\lambda$ is the 2D coordinate after camera calibration. Expanding Equation (14) yields Equations (15)–(17):

$$\lambda_u = f_x x_W + \gamma y_W + c_x z_W \tag{15}$$

$$\lambda_v = f_x y_W + c_x z_W \tag{16}$$

$$\lambda = z_W = \text{D} \tag{17}$$

where $\lambda_u$ is the x coordinate in the image; $\lambda_v$ is the y coordinate in the image; $\lambda$ is the *z*-axis coordinate of the object similar to D, which is the distance between the lens and the

object. As the distortion rate is 0, the 3D coordinates of the object can be calculated using Equations (18)–(20):

$$x_W = \frac{(u - c_x)}{f_x} D \tag{18}$$

$$y_W = \frac{(u - c_y)}{f_y} D \tag{19}$$

$$Z_W = D \tag{20}$$

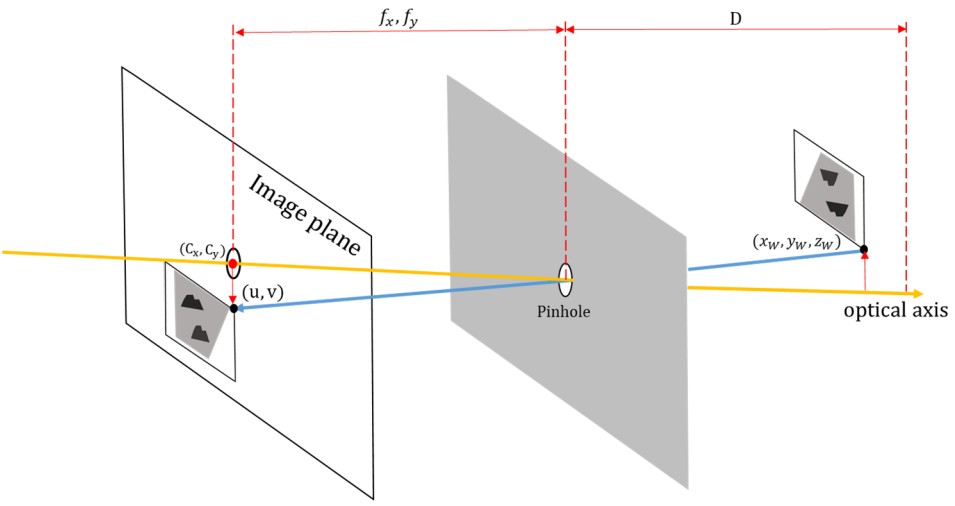

**Figure 6.** Pinhole camera model.

Using this pinhole camera formula, we attempted to convert the detected pothole size into its actual size. By substituting the D values obtained from the distance estimation equation in Equation (14) into Equations (18)–(20), the edge coordinate value of the predicted bounding box of the detected pothole can be converted into 3D coordinates. Therefore, the width and height were calculated by calculating the difference between the values of the top-left corner (x, y) and right-bottom corner (x, y) of the bounding box converted into 3D coordinate values.

## 4. Results

### 4.1. Object Detection Algorithm Performance Comparison

Deep learning model development and training were performed using the Darknet repository, a neural network framework developed by Joseph Redmond [44]. The computer environment was Google Colab Pro, which provides high-performance computing specifications with Tesla K80 with 12GB GPU memory. Table 4 lists the library environments such as CUDA, CUDNN, and TensorFlow.

**Table 4.** Training hardware environment.

| Environment | Model Version |
|:---:|:---:|
| CPU | Intel(R) Xeon(R) CPU @ 2.30 GHZ |
| GPU | Tesla K80 (12 GB) |
| IDE | Google Colab Pro |
| CUDA | 10.1 |
| Tensorflow | 2.2.1 |
| Python | 3.8 |

The learning parameters were set to a learning rate of 0.9 momentum, decay of 0.0005, batch size of 32, and a subdivision of 16. The input image size was fixed at 416 × 416, and the mix-up augmentation during the training phase was set to True so that input image

size could be randomly changed during the training phase. We set the confidence value as 0.25 and the IoU threshold as 0.5 for the proposed SPFPN-YOLOv4 tiny, and other YOLO models to detect potholes accurately. Comparing the training mAP for each model as shown in Table 5, it was confirmed that SPFPN-YOLOv4 tiny achieved not only the highest mAP(IoU = 0.5) but also showed 5.8 h of training, which is shorter than that of YOLOv2 and YOLOv3.

**Table 5.** mAP(IoU = 0.5) comparison with various YOLO models.

| Model | mAP(IoU@0.5) | Training Hours |
|---|---|---|
| YOLOv2 | 74.8% | 8.8 |
| YOLOv3 | 77.8% | 9.4 |
| YOLOv4 tiny | 72.7% | 4.2 |
| SPFPN-YOLO v4 tiny | 79.6% | 5.8 |

Figures 7–10 shows the mAP(IoU = 0.5) of YOLOv2, YOLOv3, YOLOv4 tiny, and the proposed SPFPN-YOLOv4 tiny and training hours during the training phase. The blue line in each figure indicates the loss, and the red line indicates the recorded mAP(IoU = 0.5) per 100 times from 1000 iterations. Figure 7 of YOLOv2 shows a rapidly converging loss from approximately 20 times, and shows that it achieves 74.8% mAP(IoU = 0.5) with a convergence of 1 after the training. Figure 8 of YOLOv3 shows a stable convergence pattern after 600 iterations and achieves 77.8% mAP(IoU = 0.5) with a convergence of almost zero at the 2000 iterations. Figure 9 of YOLOv4 tiny shows a stable loss pattern and converges to almost zero at 2000 iterations. Figure 7 of the proposed SPFPN-YOLOv4 tiny shows an irregular loss pattern compared to other YOLO models but achieves the highest mAP(IoU = 0.5). The training epochs of each model were set to 2000, and additional detection performance comparisons, such as precision, recall, F1-score, and FPS, were performed at the end of training.

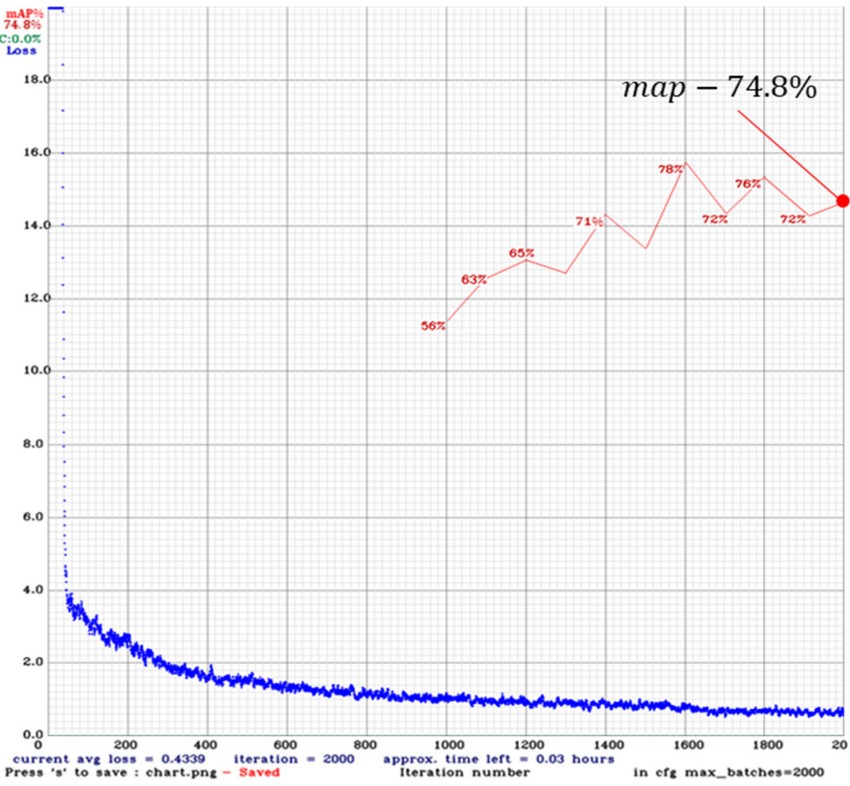

**Figure 7.** YOLOv2 training graph.

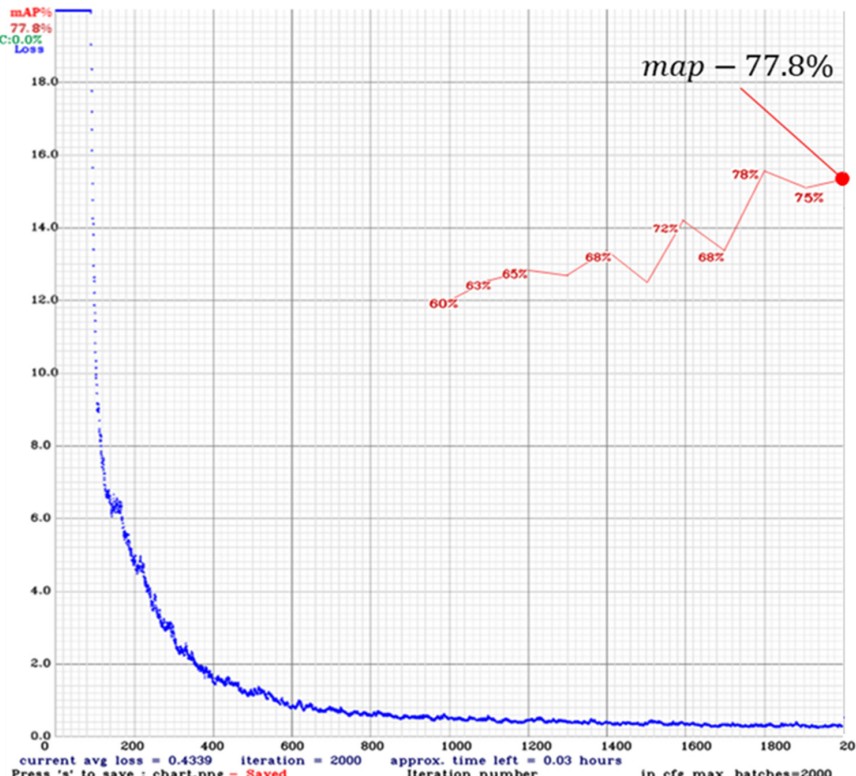

**Figure 8.** YOLOv3 training gragh.

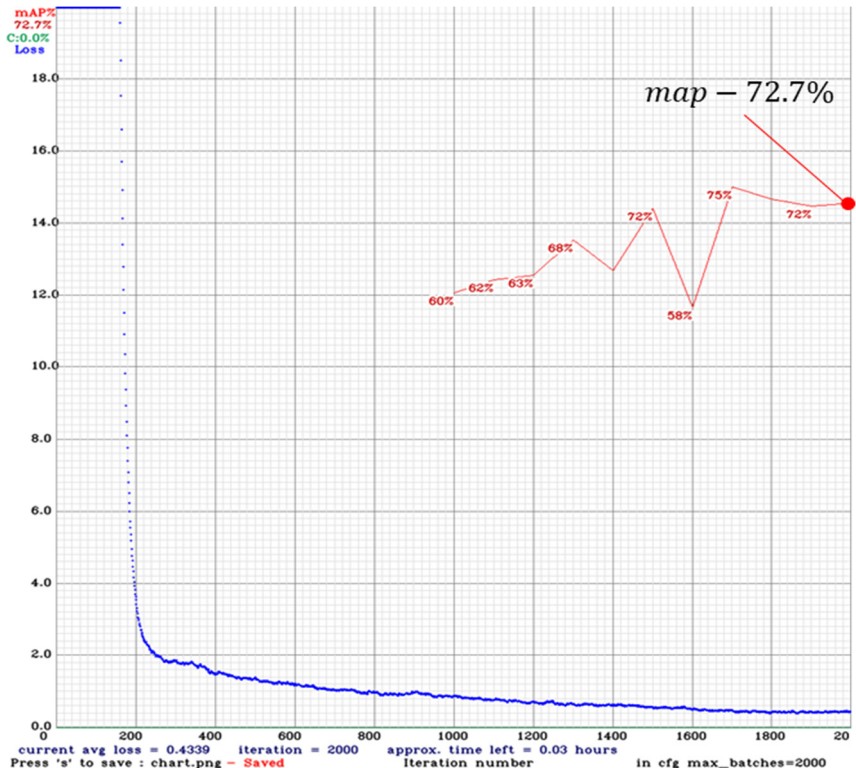

**Figure 9.** YOLOv4 tiny training gragh.

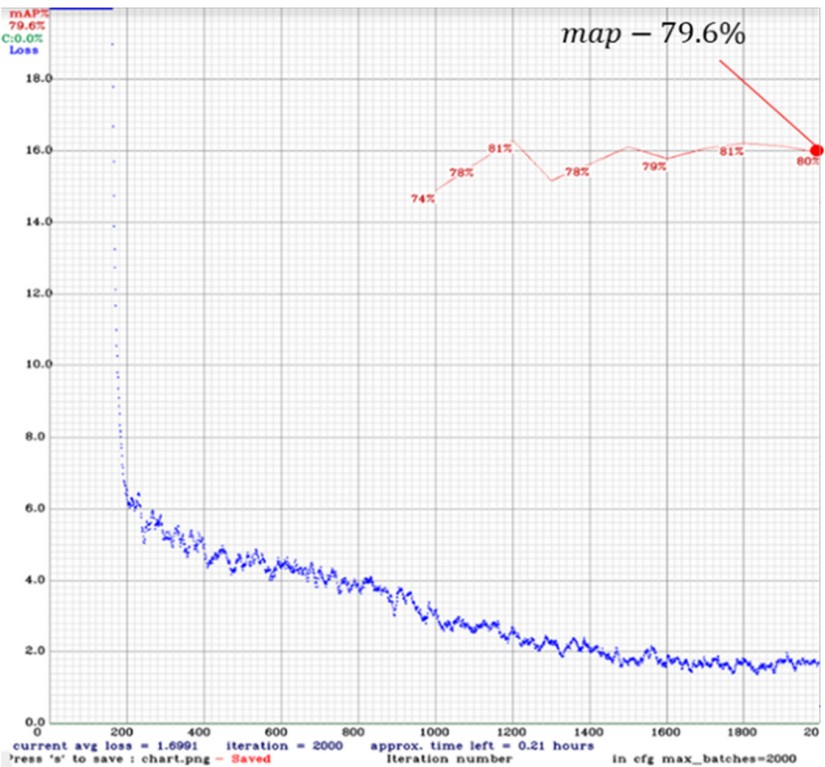

**Figure 10.** SPFPN-YOLOv4 tiny training graph.

Additionally, because the difficulty of pothole detection stems from the appearance of the model not trained on an actual road, an evaluation of the test dataset was also performed. Fifty images of the test dataset were manually selected to simulate potholes on the highway asphalt road surface. Moreover, the testing detection performance of the test dataset was compared using only a CPU (Intel i7-10700k CPU@3.80GHZ) without a GPU or GPU accelerator to assume embedded systems. Table 6 shows the precision, recall, F1-score, and FPS of each model for 50 images that are manually picked. According to the results, the proposed SPFPN-YOLOv4 tiny recorded the highest precision, recall, and F1-score. In addition, SPFPN-YOLOv4 tiny slightly exceeded the real-time detection standard of 30 FPS by achieving 38 FPS.

**Table 6.** Performance comparison on the pothole test dataset.

| Model | Precision | Recall | F1-Score | FPS |
| --- | --- | --- | --- | --- |
| YOLOv2 | 83% | 74% | 78.2% | 26 |
| YOLOv3 | 88% | 72% | 79.2% | 28 |
| YOLOv4 tiny | 74% | 76% | 72.5% | 56 |
| Proposed SPFPN YOLOv4 tiny | 89% | 84% | 86.4% | 38 |

Figure 11 shows an example of the results for the test dataset of the proposed SPFPN-YOLOv4 tiny, YOLOv4 tiny, YOLOv2, and YOLOv3. The green rectangle is the ground truth, and red is the prediction bounding box predicted by each YOLO model. The ground truths presented in pictures (a), (b), and (c) are 3, 5, and 3, respectively. In the image (a), SPFPN-YOLOv4 tiny successfully detected all three potholes, YOLOv4 tiny detected one pothole, and YOLOv2 and YOLOv3 detected two potholes. In the image (b), SPFPN-YOLOv4 tiny, YOLOv4 tiny, YOLOv2, and YOLOv3 detect four, three, two, and three potholes, respectively. In the image (C), SPFPN-YOLOv4 tiny detected all three potholes, YOLOv4 tiny and YOLOv2 detected only one pothole, and YOLOv3 detected two potholes. The values for the IoU score achieved by each YOLO model for pictures (a), (b), and (c) are summarized in Table 7.

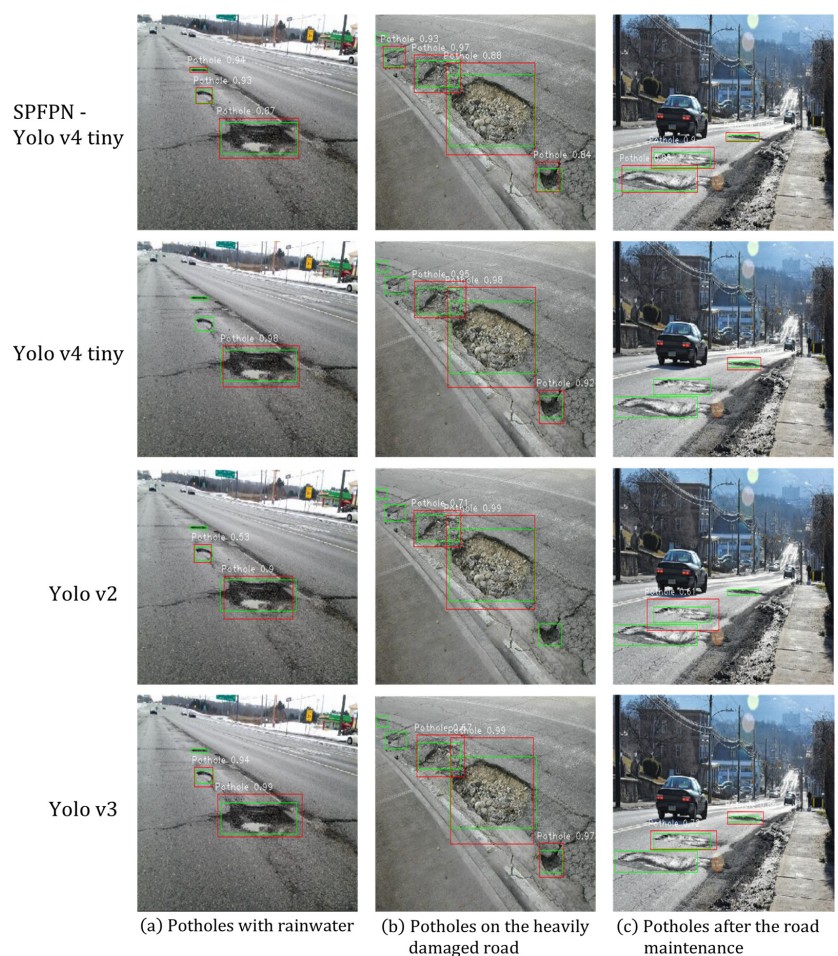

| | (a) Potholes with rainwater | (b) Potholes on the heavily damaged road | (c) Potholes after the road maintenance |

**Figure 11.** Detection result sample on the test dataset.

**Table 7.** IoU comparison according to each YOLO models.

| Picture | Model | No. of Potholes | Detected Potholes | Accuracy(%) | IoU |
|---------|-------|-----------------|-------------------|-------------|-----|
| (a) | YOLOv2 | 3 | 2 | 66 | 0.54 |
| | YOLOv3 | | 2 | 66 | 0.52 |
| | YOLOv4 tiny | | 1 | 33 | 0.61 |
| | Proposed SPFPN-YOLOv4 tiny | | 3 | 100 | 0.63 |
| (b) | YOLOv2 | 5 | 2 | 40 | 0.48 |
| | YOLOv3 | | 3 | 60 | 0.44 |
| | YOLOv4 tiny | | 3 | 60 | 0.47 |
| | Proposed SPFPN-YOLOv4 tiny | | 4 | 80 | 0.53 |
| (c) | YOLOv2 | 3 | 1 | 33 | 0.45 |
| | YOLOv3 | | 2 | 66 | 0.63 |
| | YOLOv4 tiny | | 1 | 33 | 0.52 |
| | Proposed SPFPN-YOLOv4 tiny | | 3 | 100 | 0.58 |

Potholes occur in a variety of shapes and characteristics owing to various factors (environment, physical stress, and deterioration of road surfaces), thus detectability under poor conditions should also be verified. This study attempts to solve this problem by applying data augmentation (horizontal flip, gamma regulation, and scaling). As shown in Figure 12, several detected pothole images representative of adverse conditions, such as noisy data, overexposed, underexposed, and poor quality of photos, were selected. Image (A) in Figure 12 shows a noisy road image and detects all three potholes with shallow depths and distinct boundaries at the center. In this image (A), the proposed SPFPN-YOLOv4 tiny achieved high confidence scores of 0.99, 0.98, and 0.96, respectively. Image (B) shows the results of underexposed and overexposed pothole detection at a high average confidence score of 0.95

in a road environment of poor quality with asphalt cracks and potholes. It verifies precise classification between cracks and potholes. Image (C) shows the number of pothole images with accumulated rainwater and strong light reflection at the center of each pothole. In the detection result of image (C), the proposed SPFPN-YOLOv4 tiny shows the detection of six potholes despite the presence of light reflection, but achieved a relatively lower confidence score than other images. Image (D) shows paved roads after snowfall without lane boundaries and an environment where pothole discrimination is difficult. The proposed SPFPN-YOLOv4 tiny shows that if the characteristics of the boundary and center of the road surfaces are shadowy, they are detected as potholes in image (D). However, if the distinction between the center of the road surface and the boundary is ambiguous, they are not detected as potholes. Image (E) shows underexposed potholes filled with rainwater at the road surface. In this image, four potholes were detected, with a confidence score of 0.81, 0.86, 0.9, and 0.72, respectively. The result shows the detectability in a relatively long distance. In addition, the road surface area, which shows a large amount of rainwater next to the white-dotted lane, is not detected as a pothole. Image (F) shows the results of the pothole detection of the proposed SPFPN-YOLOv4 tiny on shadowed unpaved roads. This result demonstrates the detectability of potholes that can be easily found on unpaved roads while verifying that they can be detected in environments with less light reflection.

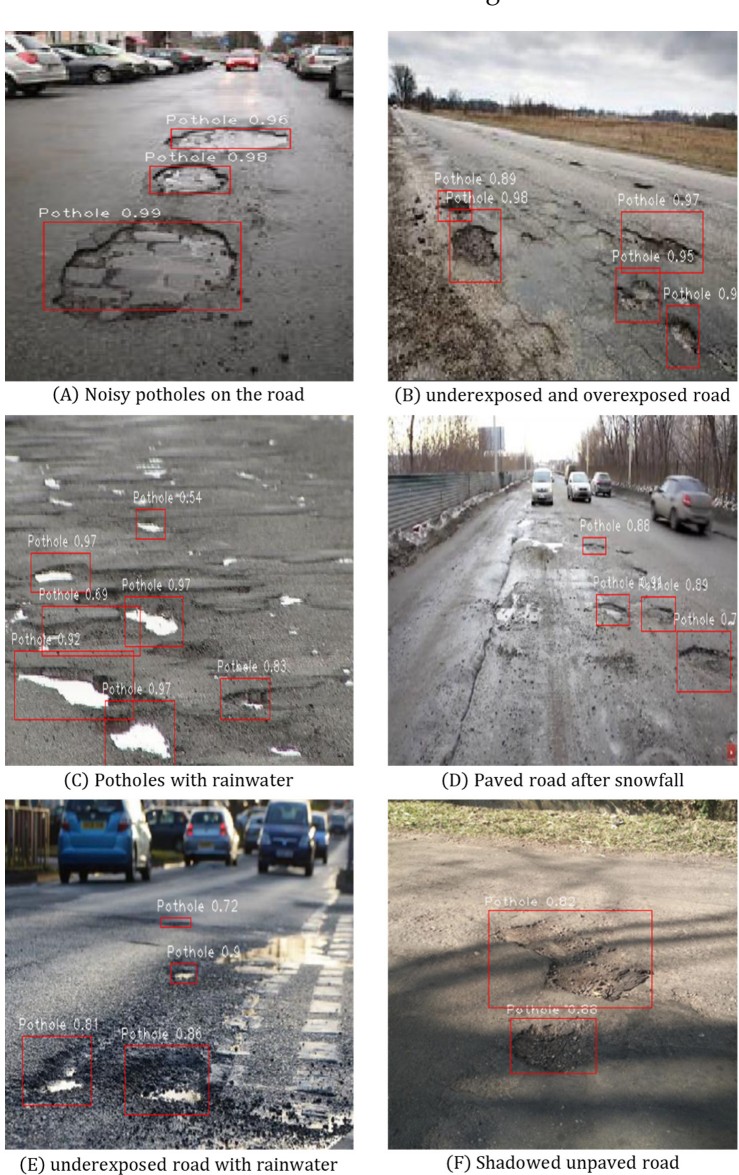

(A) Noisy potholes on the road      (B) underexposed and overexposed road

(C) Potholes with rainwater      (D) Paved road after snowfall

(E) underexposed road with rainwater      (F) Shadowed unpaved road

**Figure 12.** Detection result in the challenging condition.

### 4.2. Risk Assessment

The three-dimensional size of the potholes was calculated using pinhole cameras and a distance equation. The variables required to compute the distance used in the pinhole camera equation are the height of the camera lens, distance to the center point, inclination angle of the camera lens, and height of the image plane. We set the height of the camera (1520 mm), the distance to the center point (2900 mm), the size of the image plane (416 × 416), and the inclination angle of the camera lens (60°).

These variables can be measured by attaching a camera to the vehicle. However, the variables were arbitrarily set to verify the validity of this study. Moreover, it is desirable to measure the tire contact patch size for comparison with a pothole; this study replaced it with the tire footprint value used in other studies to verify its validity. The size of the tire contact patch is defined as 100 mm (width) × 280 mm (height) when the air pressure is 1.2 bar, and the weight under the axis load is 4383 N. Based on these values, the width and height of the tire contact patch were compared with those of the pothole width and height. It is defined as dangerous when the size of the pothole is 1.2 times larger than the tire patch size, cautious when the two values are the same, and safe when the pothole size is smaller than the tire contact patch size. Figure 13 shows the process of determining the risk using a pothole-report video. After detecting the pothole, the width and height values of the bounding box are calculated and compared with the predefined tire contact patch value so that the safety case can be checked in green and the dangerous case in red.

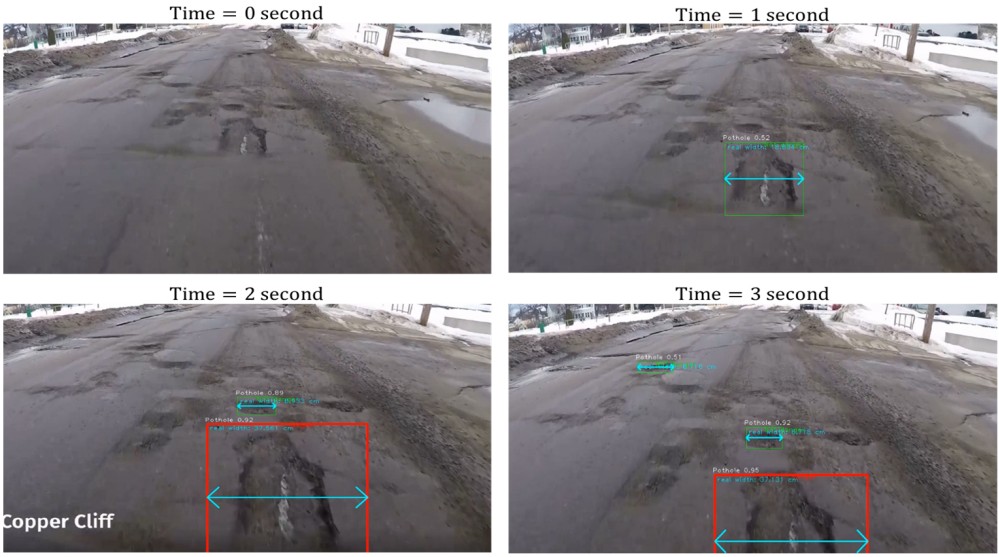

**Figure 13.** Risk assessment through detected pothole using SPFPN-YOLOv4 tiny.

## 5. Discussion

Potholes are challenging to detect because of their uneven shape owing to various factors, and they should be detected quickly, even in embedded systems. In this study, a multi-scale detection network was combined with CSPDarknet53-tiny, the backbone of YOLOv4 tiny, so that the characteristics of the object could be richly extracted. As a result, the highest mAP(IoU = 0.5) compared to YOLOv2, YOLOv3, and YOLOv4 tiny was achieved in the training phase. The highest precision, recall, and F1-score were also recorded for the fifty images manually picked in the test datasets. As a result of measuring the reference speed without utilizing the GPU to assume an embedded system, 38 FPS was achieved, exceeding the real-time detection criterion of 30 FPS. In Figure 11, which shows the inference results of each model for the test data, we showed that the proposed SPFPN-YOLOv4 detected tiny potholes with an IoU score of more than 0.5 for all images. In addition, Figure 12 shows that detection under poor academic conditions, such as noisy, shadowy, poor-quality, underexposed, and overexposed, is possible.

In addition, this study developed a pothole risk assessment algorithm based on 2D images that have yet to be addressed in previous studies, focusing on the fact that potholes should be evaluated differently in their risk depending on their size. If the risk can be evaluated in real-time, it can not only be used for road maintenance, but also for detecting large potholes that are not recognizable by drivers in driving situations.

## 6. Conclusions

For embedded systems with a low-grade computing performance, a new 2D pothole detection algorithm was proposed. SPFPN-YOLOv4 tiny was developed by combining CSPDarknet53-tiny with SPP and FPN to extract rich features from images.

Data augmentation techniques, such as gamma regulation, horizontal flip, and scaling, were applied to compensate for insufficient datasets. The height and width of the anchor box were calculated by applying the K-means++ algorithm to optimize the IoU score. After applying data augmentation, 2665 image datasets were obtained, which were then divided into training, validation, and testing data at 70%, 20%, and 10% ratios, respectively. A total of 2400 images were used for the training and validation phases, and 265 images were used to evaluate the detection performance of the various YOLO models on the test dataset. As a result of the training, SPFPN-YOLOv4 tiny, YOLOv4 tiny, YOLO 2, and YOLOv3 models achieved 79.6%, 72.7%, 74.8%, and 77.8%, respectively. Subsequently, 50 pothole images were manually selected from the test dataset to reflect the highway asphalt pavement environment. On measuring precision, recall, and F1-score, the Proposed SPFPN-YOLOv4 tiny showed the highest values. In addition, real-time detection was verified by achieving 38 FPS as a result of inferring images using only a CPU(Intel i7-10700k CPU@3.80GHZ).

In conclusion, we developed a lightweight pothole detection algorithm that enables real-time detection, even in an environment without a GPU, and that improves accuracy over general YOLO series, such as YOLOv2 and YOLOv3. In addition, we proposed a novel risk assessment algorithm that predicts the actual size of large potholes that are likely to have a negative impact on the vehicle. Moreover, it can determine the risk before a vehicle comes into contact with the pothole by comparing it with the tire contact patch size. Thus, this algorithm can not only be applied to road maintenance in various countries evaluating the risk considering pothole size but also help the driver discover largely unrecognized potholes in advance. As the number of pothole-related accidents is rapidly increasing, continuous attention and response are required to mitigate them. Therefore, if the pothole detection algorithm developed in this study is applied, the number of accidents resulting from potholes can be reduced. In future research, it will be necessary to verify real-time detection in an actual road environment by attaching an embedded system to the vehicle. In this research, although dangerous potholes are marked as red bounding boxes so that they can only be visually identified, the signal processing method will be combined with our new risk assessment to indicate risk signals, such as vibration and haptic signals that the driver can feel later.

**Author Contributions:** Conceptualization, T.-O.T.; Methodology, D.-H.H.; Software, D.-H.H.; Validation, D.-H.H.; Resources, T.-O.T.; Data Curation, J.-Y.C. and S.-B.K.; Writing—Original Draft Preparation, D.-H.H.; Writing—Review & Editing, S.-P.Z.; Visualization, D.-H.H. and S.-P.Z.; Supervision, S.-P.Z.; Project Administration, S.-P.Z.; Funding Acquisition, T.-O.T.; Investigation, D.-H.H. All authors have read and agreed to the published version of the manuscript.

**Funding:** This work was supported by the Ministry of Trade, Industry, and Energy (20011729).

**Institutional Review Board Statement:** Not applicable.

**Informed Consent Statement:** Not applicable.

**Data Availability Statement:** The data reported in this study are included in the article.

**Acknowledgments:** The authors would like to thank Yongjun Pan, a mechanical and vehicle engineering professor at Chongqing university, for his constructive comments and extensive English review and writing, which significantly strengthened this paper. This work was supported by the Korea Institute of Industrial Technology Evaluation and Management grant funded by the Korea government (Ministry of Trade, Industry and Energy) in 2022. (No.20011729, Development of International Standard for Pedestrian Collision Prevention System Performance Evaluation Technology for Autonomous Vehicles).

**Conflicts of Interest:** The authors declare no conflict of interest.

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
