# Peer review of "Image-Based Pothole Detection Using Multi-Scale Feature Network and Risk Assessment"

_electronics, doi:10.3390/electronics12040826_

Round 1
Reviewer 1 Report
The topic of the paper is interesting to the readers of the field. The paper is well organized and written. However, authors are suggested to add some comparison results with other methods to verify the advantage of your proposed method.
Reviewer 2 Report
Image-based pothole detection using multi-scale feature 3 network and risk assessment
The paper discusses an interesting problem. However, I have following concerns on the paper.
· In the abstract please discuss about the dataset used in this study.
· In the introduction can you give statistics on year 2022 instead of 2020 at reference 1.
· Introduction should be enlarged. Only 4 references in this section. Try to have atleast 10-12 references from 2021 and 2022.
· Make a separate paragraph called Major significant contribution and list the contributions in bullet form at the end of introduction.
· Make a paragraph about the organization of the paper at the end of introduction.
· Literature review section should table at the end summarizing the whole section.
· Please mention what is the research gap.
· Make a separate section for dataset and put all the information related to dataset, statistics, graph figure to show the number of files, pre-processing, data augmentation.
· Before 3.2.1 please give some introductory about this section.
· What are the performance evaluation metrics?? Please explain them.
· Please write a discussion section
· Make a separate section for comparative analysis and give proper references with whom the comparison is done.
· Results section is very week. Support the results with suitable graphs and figure.
· Confusion metrics, Accuracy and Loss graphs missing.
Reviewer 3 Report
Dear Authors,
I have read your paper with attention and pleasure.
In my opinion, the manuscript titled Image-based pothole detection using multi-scale featurenetwork and risk assessment presents original research and innovative solutions and could be interesting for readers of the MDPI Electronics Journal. n my opinion, it also corresponds to the theme of Deep Perception in Autonomous Driving Special Issue. The article deals with very practical and important issues from the point of view of road users regarding pothole detection. an innovative approach based on a multi-scale feature network was used, and risk assessment was also taken into account.
The motivation is clear. The object of study, as well as the results, are comprehensively described providing valuable conclusions.
The paper is organised in a logical manner. The state of the art covers the main results in the field. The contributions of the report are clearly stated in the Introduction chapter.
I have no objections to recommending publishing this paper. However, due to the listed below drawbacks, my recommendation is "Accept after minor revision". In my opinion, several aspects require clarification. Please revise and add some comments and improvements according to the following:
- in my opinion, an analysis of the effectiveness of the proposed algorithm should be performed for noisy data, i.e. a base of patterns in the form of poor-quality photos, with light reflections, underexposed, overexposed, etc. This will undoubtedly increase the quality of the article, prove the effectiveness of the method, also for imperfect data sets,
- - the innovativeness of the presented new image-based risk assessment algorithm based on the relationship between the impact on the vehicle and the size of the pothole should be demonstrated more strongly,
- several typos errors should be removed, for example, the titles of Fig. 4 and 5 should start with a capital letter, after recalling the number of the figure, no point should be placed ("Figure 4." is present, "Fig. 4" should be) - see lines 318, 346, 412, 447.
Round 2
Reviewer 2 Report
The authors have revised the paper based on the comments.
